# Photo-induced intramolecular dearomative [5 + 4] cycloaddition of arenes for the construction of highly strained medium-sized-rings

Min Zhu [1,2,3], Yuan-Jun Gao[1,3], Xu-Lun Huang[1,2,3], Muzi Li[1], Chao Zheng [1] ✉ & Shu-Li You [1,2] ✉

Medium-sized-ring compounds have been recognized as challenging synthetic targets in organic chemistry. Especially, the difficulty of synthesis will be augmented if an *E*-olefin moiety is embedded. Recently, photo-induced dearomative cycloaddition reactions that proceed via energy transfer mechanism have witnessed significant developments and provided powerful methods for the organic transformations that are not easily realized under thermal conditions. Herein, we report an intramolecular dearomative [5 + 4] cycloaddition of naphthalene-derived vinylcyclopropanes under visible-light irradiation and a proper triplet photosensitizer. The reaction affords dearomatized polycyclic molecules possessing a nine-membered-ring with an *E*-olefin moiety in good yields (up to 86%) and stereoselectivity (up to 8.8/1 *E/Z*). Detailed computational studies reveal the origin behind the favorable formation of the thermodynamically less stable isomers. Diverse derivations of the dearomatized products have also been demonstrated.

It is well known that the synthesis of medium-sized-ring compounds ranks among challenging goals in organic chemistry due to unfavorable transannular interactions and entropy effect[1–4]. In planning a synthesis towards a target molecule having a medium-sized-ring structure, only limited strategies, such as ring-closing metathesis[5–9], sigmatropic rearrangement and related ring-expansion from small-ring precursors[10–14], and multi-component cyclization[15–17], etc., are typically employed to fulfill the task of ring-closure. Particularly, the synthetic difficulty will be further increased if an *E*-olefin moiety is embedded in the medium-sized-ring. Compared with the *Z*-olefin, the connection of the *E*-olefin with the remaining part of the medium-sized-ring through the two bonds at the opposite side of a C = C double bond imposes significant inherent rigidity of the ring structure as well as the strong ring strain, which impedes the facile synthesis of natural products and bioactive molecules possessing such structural units (Fig. 1a)[18–20]. Traditional methods to access these structures mainly relied on intramolecular cyclization or the functional group manipulations on a pre-assembled medium-sized-ring (such as elimination, semi-hydrogenation, and photoinduced isomerization, etc.), which usually suffered from low reactivity and selectivity issues[21,22]. To be noted, Cheng and Wagner reported in 1994 an intramolecular cycloaddition of cyclopropyl-substituted *p*–(butenyloxy)acetophenone under the irradiation of ultraviolet (UV) light. The [5 + 4] adduct possessing an *E*-olefin was identified as a major product in the reaction mixture but not isolable due to its instability (Fig. 1b)[23]. In this regard, the development of efficient methods for the construction of medium-sized-ring molecular scaffolds with an *E*-olefin embedded is highly desirable but synthetically challenging.

[1]New Cornerstone Science Laboratory, State Key Laboratory of Organometallic Chemistry, Shanghai Institute of Organic Chemistry, University of Chinese Academy of Sciences, Chinese Academy of Sciences, 345 Lingling Lu, Shanghai, China. [2]School of Physical Science and Technology, ShanghaiTech University, 100 Haike Road, Shanghai, China. [3]These authors contributed equally: Min Zhu, Yuan-Jun Gao, Xu-Lun Huang. ✉e-mail: zhengchao@sioc.ac.cn; slyou@sioc.ac.cn

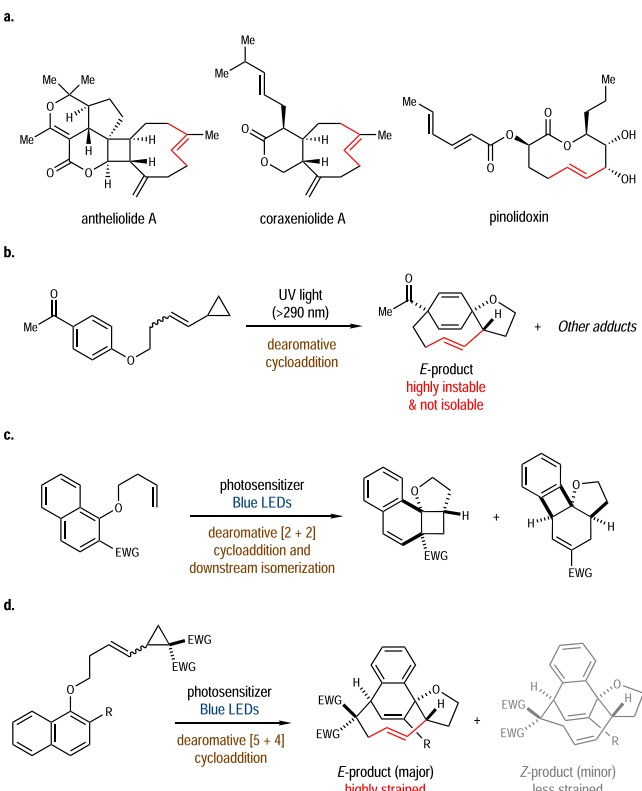

**Fig. 1 | The synthesis of medium-sized-ring molecules containing Z/E-olefins via dearomative cycloaddition reactions. a.** Selected natural products with medium-sized-rings containing an E-olefin. **b.** UV promoted dearomative cycloadditions of benzene-derived VCP[23]. **c.** Energy transfer enabled dearomative cycloadditions of naphthalene-derives[50]. **d.** Synthesis of medium-sized-rings containing an E-olefin via dearomative cycloaddition (This work).

Visible-light-induced dearomative cycloaddition reactions of heteroaromatics with various unsaturated functionalities via Dexter energy transfer mechanism[24–28] have recently emerged as a powerful method for the synthesis of three-dimensional polycyclic molecules with complex topologies[29,30]. Among diverse types of arene substrates that are suitable for this reaction manifold, indoles/pyrroles are most studied. Intra- or intermolecular [$m+2$], or [$m+4$] cycloaddition reactions with (functional) alkenes, alkynes, allenes, oximes, (hetero) arenes or vinylcyclopropanes (VCPs) have been reported[31–39]. Various chemo-, regio-, diastereo- and enantioselectivity issues can be well regulated. In this regard, our group reported in 2021 an intramolecular dearomative [$5+2$] cycloaddition of indole-tethered VCPs, affording polycyclic indolines bearing a seven-membered-ring possessing a Z-olefin moiety[39]. Mechanistic studies revealed that the ring-opening/-closing of the VCP moiety are in equilibrium during the reaction sequence, thus, both Z- and E-olefin intermediates were generated efficiently. However, due to the geometric restriction in the formation of a seven-membered-ring, only Z-olefin can be conserved in the final products.

The dearomative cycloaddition reactions of electronically unbiased arenes are more challenging than those of heteroarenes like indoles, because excitation of unbiased π-electrons required much more energy inputs[40–42]. Traditional reactions under UV conditions were synthetically less useful due to often-uncontrollable selectivity[43–45]. Notably, a series of enabling dearomative transformations of naphthalenes/benzenes have been reported, which were initiated with visible-light-induced cycloaddition with arenophiles like 4-methyl-1,2,4-triazoline-3,5-dione (MTAD) by Sarlah[46–48] and 1,2-dihydro-1,2,4,5-tetrazine-3,6-diones (TETRADs) by Yoshino and

Matsunaga[49]. The groups of Glorius and others achieved a series of photo-induced dearomative [$2+2$] and related cycloadditions of naphthalenes (Fig. 1c) or bicyclic (aza)arenes with alkenes or bicyclo[1.1.0]butanes[50–62]. As a further step forward to our long-term goal of catalytic dearomatizations[63–69], we recently realized the photo-induced dearomative [$5+4$] cycloaddition of naphthalene-derived VCPs (Fig. 1d). Unlike the reaction with the indole-based starting materials, this reaction dominantly delivered the dearomatized products possessing an E-olefin moiety embedded in a nine-membered-ring (E/Z ratio up to 8.8/1). The reaction provides a unique entry into highly strained medium-sized-ring compounds and allows further increases of molecular complexity by diverse transformations of the [$5+4$] cycloadducts. Combined experimental and computational studies demonstrate the origin of this uncommon synthesis of highly strained medium-sized-ring compounds. Herein, we report the results from this study.

## Results

### Reaction development

Our study began with the evaluation of the reaction conditions of the dearomative cycloaddition with naphthalene-derived VCP (**1a**, 4/1 E/Z) as the substrate (Table 1). A series of commonly used triplet photosensitizers shown in Fig. 2 was employed in the presence of photo-irradiation (24 W, 455 nm) with DCM as the solvent at room temperature. No reaction was observed with Rose Bengal (**I**) or Ru(bpy)$_3$(PF$_6$)$_2$ (**II**) which have relatively small energy gaps between the lowest triplet (T$_1$) and singlet (S$_0$) states [$\Delta G$(T$_1$–S$_0$) = 40.9–46.5 kcal mol$^{-1}$] (entries 1–2). With the Ir-based photosensitizers (**III–VI**) whose triplet–singlet energy gaps are higher [$\Delta G$(T$_1$–S$_0$) = 49.2–63.5 kcal mol$^{-1}$], the dearomative [$5+4$] cycloaddition occurred smoothly, affording the corresponding bridged cyclic products E-**2a** and Z-**2a** with a nine-membered-ring embedding either an E- or Z-olefin moiety (entries 3–6). The structures of E-**2a** and Z-**2a** have been established unambiguously by X-ray crystallographic analysis. As expected, significant deviation from the coplanarity of the olefin moiety was observed in E-**2a** but not in Z-**2a** (the dihedral angles D(C$^a$–C$^b$–C$^c$–C$^d$) in the two molecules are 143.1° and 5.1°, respectively). Particularly, the optimal yields of E-**2a** (86% NMR yield and 85% isolated yield) and the E/Z ratio of **2a** (7.7/1) were obtained when Ir(ppy)$_3$ (**IV**) was employed (entry 4). The lower E/Z ratios (less than 3/1, entries 5 and 6) with photosensitizers **V** or **VI** were attributed to the conversion of E-**2a** to Z-**2a** along with further [$2+2$] cycloadducts **7** and **8** (vide infra) under these conditions. The effects of various solvents were also considered (entries 7–12). The reactions proceeded well in acetone, MeCN, dioxane, DMSO and EtOAc, delivering E-**2a** in comparable yields (71–82%) and E/Z ratios (5.4–7.4/1). However, the utilization of MeOH completely inhibited the reaction (entry 8). It should be noted that except for the [$5+4$] cycloaddition, other known patterns of the cycloaddition involving VCP or naphthalenes, like [$2+2$] or [$5+2$], were not observed in all these cases. Control experiments confirmed that both photoirradiation and the presence of a photosensitizer were necessary for promoting the desired reaction (entries 13 and 14).

With the optimized reaction conditions in hand (entry 4, Table 1), the substrate scope of the dearomative [$5+4$] cycloaddition was investigated (Fig. 3). Variations of the R$^1$ group from acetyl to different alkyl ketone moieties and methoxycarbonyl allowed the facile formation of the dearomatized medium-sized-ring products (**2b–2f**, 38–80% yields, 6.7–8.6/1 E/Z). The substituents on the VCP group (R$^3$) did not affect the reaction outcomes in that the corresponding products **2g–2i** were afforded smoothly in 70–80% yields with 6.3–8.8/1 E/Z. Notably, even unsubstituted VCP (R$^3$ = H) could participate in the desired reaction (**2j**, 70% yield), albeit with decreased E/Z ratio (2.6/1). Meanwhile, the current method has a good tolerance of the naphthalene derivatives bearing diverse substituents on the C5 or C6 position (R$^2$ = F, Cl, Br, I, Ph, phenylethynyl,

**Table 1 | Optimization of the Reaction Conditions[a]**

| entry | PS | solvent | time (h) | yield of E-2a (%)[b], E/Z ratio[c] |
|---|---|---|---|---|
| 1 | I | DCM | 48 | 0, / |
| 2 | II | DCM | 24 | 0, / |
| 3 | III | DCM | 24 | 84, 5.4/1 |
| 4 | IV | DCM | 24 | 86 (85[d]), 7.7/1 |
| 5 | V | DCM | 24 | 69, 2.8/1 |
| 6 | VI | DCM | 24 | 72, 2.9/1 |
| 7 | IV | acetone | 24 | 72, 5.4/1 |
| 8 | IV | MeOH | 24 | 0, / |
| 9 | IV | MeCN | 24 | 82, 6.4/1 |
| 10 | IV | dioxane | 24 | 78, 6.2/1 |
| 11 | IV | DMSO | 24 | 71, 7.1/1 |
| 12 | IV | EtOAc | 24 | 76, 7.4/1 |
| 13[e] | / | DCM | 36 | 0, / |
| 14[f] | IV | DCM | 36 | 0, / |

[a] Reaction conditions: **1a** (0.1 mmol) and photosensitizer (PS, 1 mol%) in solvent (1 mL) were irradiated by blue LEDs (24 W, 455 nm) at room temperature under argon for a specified time.

[b] $^1$H NMR yield of E-**2a** using $C_2H_2Cl_4$ as an internal standard.

[c] E/Z ratio determined by $^1$H NMR spectrum of the crude reaction mixture.

[d] Isolated yield of E-**2a**.

[e] In the absence of a PS.

[f] In dark.

methoxy and Bpin). All the dearomative [5 + 4] adducts **2l**–**2t** were delivered in moderate to good yields (54–82%) and E/Z ratios (4.8–8.2/1). Particularly, the introduction of halogen atoms or a Bpin group to the bridged cyclic products offered flexible handles for further elaboration of molecular complexity. As a demonstration, the dearomative [5 + 4] cycloaddition could be performed at 3.9 mmol scale, delivering E-**2a** in 75% yield (7.3/1 E/Z, 1.15 g). Notably, if the $R^1$ group was switched to methyl or H, the desired reaction was not observed with the starting materials recovered. According to our previous works[26,32], adding an electron-withdrawing group would be beneficial for the energy-transfer-mediated dearomative cycloaddition. Some unsuccessful substrates for this dearomative [5 + 4] cycloaddition reaction are listed in the Supplementary Fig. 9.

On the basis of our previous works and literature reports, the current dearomative [5 + 4] cycloaddition reaction is believed to proceed via a Dexter energy transfer mechanism[21,22]. Accordingly, the yield of the reaction from **1a** to E-**2a** was lowered to 13% when the reaction was performed in the presence of 2,5-dimethylhexa-2,4-diene (1 equiv), a known triplet quencher[70]. Detailed DFT calculations have been

performed in order to provide the energy landscape of the reaction (Fig. 4). In the presence of blue LED irradiation and photosensitizer **IV**, **1a** could be excited into the $T_1$ state, with the increase of Gibbs free energy of 59.7 kcal mol$^{-1}$ (**1a-T$_1$**). The unpaired electrons were mainly distributed around the substituted phenyl ring, with the notable Mulliken spin density population on the C1 (0.568), C2 (0.376) and C4 (0.463) positions. The radical attack of the C1 position of the naphthalene ring to the VCP moiety was quite facile, associated with a small energetic barrier on the $T_1$ state (**TS1-T$_1$**, 65.5 kcal mol$^{-1}$). In the following biradical intermediate **INT1-T$_1$** (50.0 kcal mol$^{-1}$), one electron was formally located in adjacent to the cyclopropane ring (0.959), and the other was roughly averagely located on the C2 (0.505) and C4 (0.521) positions of the naphthalene ring. The next step was the ring-opening of the cyclopropane ring along with the formation of an E- or Z-olefin when the two hydrogen atoms were located at the opposite or the same side of the newly generated C = C double bond in the corresponding transition states (**TS2-E-T$_1$**, 53.4 kcal mol$^{-1}$, **TS2-Z-T$_1$**, 54.3 kcal mol$^{-1}$). The structures, energies and spin-states ($T_1$ or open-shell singlet, OSS) of the subsequent biradicals possessing either

an *E*-or *Z*-olefin moiety were all determined (**INT2-*E*-T$_1$**, 44.9 kcal mol$^{-1}$, **INT2-*Z*-T$_1$**, 45.1 kcal mol$^{-1}$, **INT2-*E*-OSS**, 45.0 kcal mol$^{-1}$ and **INT2-*Z*-OSS**, 45.8 kcal mol$^{-1}$), so were the corresponding minimal energy crossing points (MECP) for the spin inversion processes (**MECP-1** and **MECP-2**) (The calculated electronic energies (at the (U)B3LYP-D3(BJ)/def2-SVP (SMD, DCM) level of theory) of **MECP-1** and **MECP-2** were 37.8 kcal mol$^{-1}$ and 40.1 kcal mol$^{-1}$, respectively, related to that of **1a**.).

In agreement with the experiments, our calculations suggested that the biradical recombination only occurs at the C4 position of the naphthalene ring (**TS3-*E*-OSS**, 46.1 kcal mol$^{-1}$ and **TS3-*Z*-OSS**, 49.1 kcal mol$^{-1}$), and the transition states for the [5 + 2] cycloaddition via the C−C bond-formation at the C2 position were not located. Starting from the ring-opening of the cyclopropane, all the downstream transition states and intermediates with an *E*-olefin moiety were lower in energy than their *Z*-counterparts. On the contrary, for the final medium-sized-ring products, the isomer embedding an *E*-olefin (*E*-**2a**, 12.9 kcal mol$^{-1}$) was thermodynamically much less stable than the isomer with a *Z*-olefin (*Z*-**2a**, 5.4 kcal mol$^{-1}$). The plausible reasons behind this phenomenon might be as follows: (a) Although the *E*-olefin in the final product experienced much stronger ring-strain, neither did the facile biradical recombination transition states (Both **TS3-*E*-OSS** and **TS3-*Z*-OSS** were quite early transition states with long distances (about 3.30 Å) of the forming C−C bonds) nor their precursors. (b) The discrimination between the two reaction pathways was originated during the formation of the *E*- or *Z*-olefin in the step of ring-opening of cyclopropane ring. The formation of *E*-olefin via the transition state **TS2-*E*-T$_1$** was favorable than **TS2-*Z*-T$_1$** that led to *Z*-olefin. The lower *E*/*Z* ratio of **2j** (2.6/1) might be attributed to the attenuated energetic difference between the transition states for ring-opening of cyclopropane ring due to the missing of steric congestion between the *gem*-diester groups and the naphthalene ring. (c) The olefinic geometry of the substrate would not affect the *E*/*Z* ratio of the product since configurational difference of the substrate would vanish after the generation of triplet 1,4-biradical via the first C−C bond-formation, while the *E*/*Z* configuration in the product would be set up during the subsequent ring-opening of cyclopropane ring.

**Fig. 2 | The photosensitizers used in this study.** The triplet energies of photosensitizers are cited from ref. 25.

**Fig. 3 | Substrate Scope.** Reaction conditions: **1** (0.2 mmol) and Ir(ppy)$_3$ (1 mol%) in DCM (2 mL) were irradiated by blue LEDs (24 W, 455 nm) at room temperature under argon for 24 h. The *E*/*Z* ratios of **2** were determined by $^1$H NMR spectrum of the crude reaction mixture. Isolated yields of *E*-**2** are reported. Note: (a) Combined yield of *E*-**2j** and *Z*-**2j**.

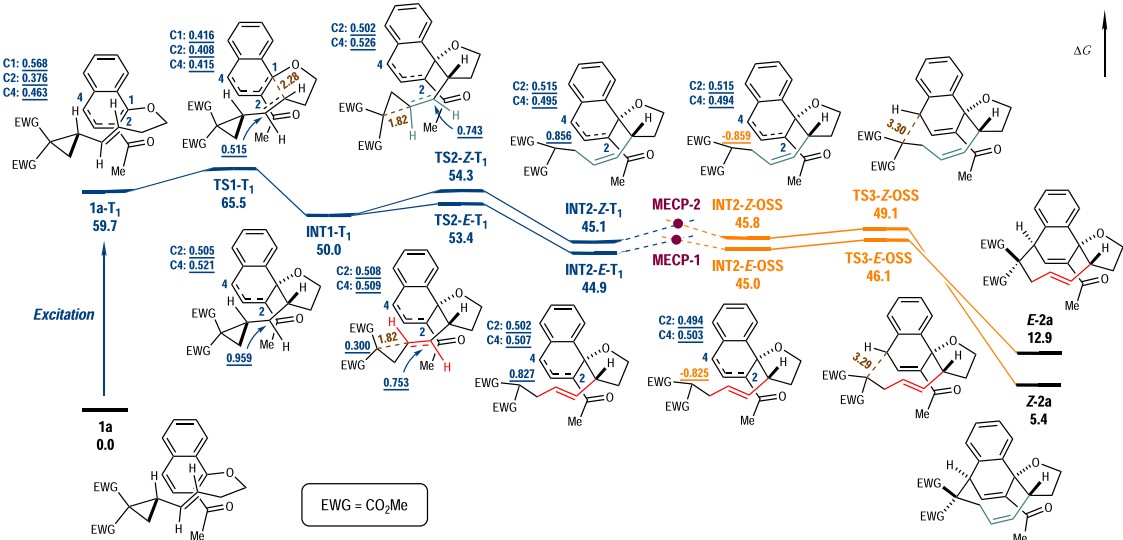

**Fig. 4 | Energy profile of the dearomative [5 + 4] cycloaddition reaction of naphthalene-derived VCP.** Calculated at the (U)M062X-D3/def2-TZVPP (SMD, DCM)//(U)B3LYP-D3(BJ)/def2-SVP (SMD, DCM) level of theory. The Gibbs free energies (ΔG) are in kcal mol⁻¹. The values in brown are bond distances (in Å), and the underlined values are Mulliken spin populations at specified atoms.

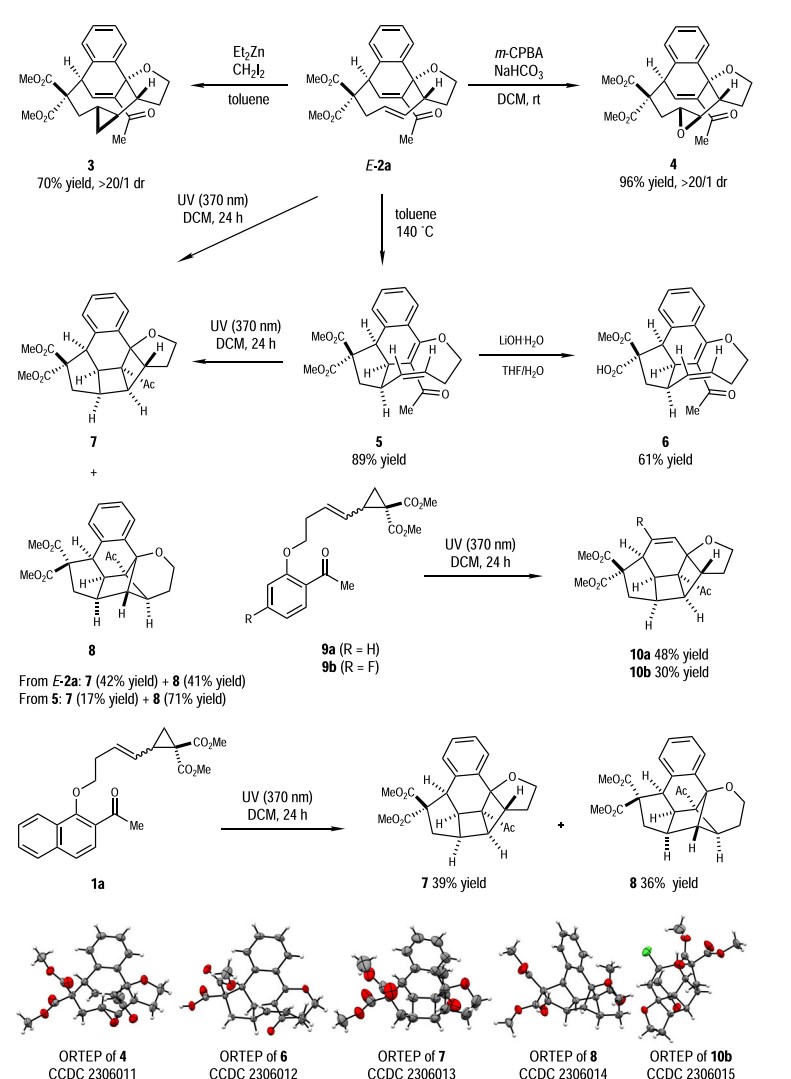

**Fig. 5 | Diverse transformations further building-up molecular complexity.**

The dearomatized [5 + 4] cycloadducts were examined for further derivations (Fig. 5). Under typical conditions of cyclopropanation, compound E-**2a** was transformed to the desired cyclopropane **3** in 70% yield. Meanwhile, the epoxidation occurred smoothly in the presence of *m*-CPBA as the oxidant, affording epoxide **4** in 96% yield. Notably, compound **4** could also be obtained in 46% yield when E-**2a** was heated at 80 °C under air. These reactions proceeded specifically at the 1,2-disubstituted C = C double bond of E-**2a** with exclusive diastereoselectivity (>20/1 dr). When heated at 140 °C, a Cope rearrangement translocated the two double bonds of E-**2a** along with the reorganization of the 5/9 ring system. The ring strain was largely released in that the newly formed 1,2-disubstituted C = C double bond in **5** (89% yield) was in the Z-configuration. Partial hydrolysis of the diester moiety of **5** with LiOH•H$_2$O provided monocarboxylic acid **6** in 61% yield. When exposed under the UV irradiation (370 nm) for 24 h, both E-**2a** and **5** were transformed into more complex structures with either an additional [2.2.0] bicycle in **7** (42% yield from E-**2a**, 17% yield from **5**) or an additional [2.1.1] bicycle in **8** (41% yield from E-**2a**, 71% yield from **5**). The conversion from **5** to **7** and **8** required one [2 + 2] cycloaddition step in either the *head-to-head* or *head-to-tail* manner. The reaction started from E-**2a** was more complicated, with photo-induced E/Z isomerization of C = C double bond and multiple [2 + 2] cycloaddition/ring-opening processes possibly involved, and Z-**2a** as an intermediate (isolated in 50% yield when the reaction was quenched at 4 h). It should be mentioned that under the irradiation of UV light (370 nm), compound **1a** could be converted to a mixture of **7** (39% yield) and **8** (36% yield), and benzene-derived VCPs **9a** and **9b** to [2.2.0] bicyclic analogs **10a** (48% yield) and **10b** (30% yield), respectively. Besides, **9a** remained intact under blue LEDs with photosensitizer **VI** or thiaxanthenone. The structures of **4, 6, 7, 8** and **10b** were determined unambiguously by X-ray crystallographic analysis.

## Discussion

We have developed a photo-induced intramolecular [5 + 4] cycloaddition of naphthalene derived VCPs. In conjunction with the dearomatization of one benzene ring, the reaction affords polycyclic molecular scaffolds containing a medium-sized-ring with a highly strained E-olefin moiety. The reactions proceed under mild conditions with excellent functional group compatibility. The desired products are obtained in moderate to good yields and E/Z ratios, and readily subjected to manipulations towards further increased molecular complexity. Detailed DFT calculations have been performed, demonstrating the energy landscape of the reaction and the origin for the favorable formation of thermodynamically less stable products.

## Methods

### Procedure for the dearomative [5 + 4] cycloadditions of naphthalene derivatives (using 1a as an example)

To a flame-dried sealed tube were added **1a** (79.0 mg, 0.2 mmol), photosensitizer **IV** (1.3 mg, 0.002 mmol), and anhydrous DCM (2.0 mL). The reaction mixture was degassed via freeze-pump-thaw for 3 cycles. After the reaction mixture was thoroughly degassed, the vial was sealed and positioned approximately 10 cm from blue LEDs (24 W, $\lambda_{max}$ = 455 nm). Then the reaction mixture was stirred at room temperature for 24 h (monitored by TLC) under argon atmosphere. Afterwards, the reaction mixture was concentrated by rotary evaporation. Then, the residue was purified by silica gel column chromatography (petroleum ether/acetone = 6/1) to afford the desired major product E-**2a** and minor product Z-**2a**.

### Procedure for the cascade dearomative [5 + 4]/[2 + 2] cycloadditions of benzene derived VCPs (using 9a as an example)

To a flame-dried sealed tube were added benzene derivative **9a** (70.0 mg, 0.2 mmol) and anhydrous DCM (20 mL). The reaction mixture was degassed via freeze-pump-thaw for 3 cycles. After the reaction

mixture was thoroughly degassed, the vial was sealed and positioned approximately 10 cm from UV LEDs (24 W, $\lambda_{max}$ = 370 nm). Then the reaction mixture was stirred at room temperature for 24 h (monitored by TLC) under argon atmosphere. Afterward, the reaction mixture was filtered and concentrated by rotary evaporation. Then, the residue was purified by silica gel column chromatography (petroleum ether/EtOAc = 3/1) to afford the desired product **10a**.

## Data availability

Detailed experimental procedures, characterization data, and computational results are provided in the Supplementary Information. Crystallographic data for the structures reported in this Article have been deposited at the Cambridge Crystallographic Data Center under deposition numbers CCDC 2306009 (E-**2a**), 2306010 (Z-**2a**), 2306011 (**4**), 2306012 (**6**), 2306013 (**7**), 2306014 (**8**), 2306015 (**10b**). Copies of the data can be obtained free of charge via https://www.ccdc.cam.ac.uk/structures/. The coordinates of all optimized structures are presented as Source Data. All other data are available from the authors upon request. Source data are provided with this paper.

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

## Acknowledgements

We thank National Key Research and Development Program of China (2021YFA1500100), National Natural Science Foundation of China (21821002, 22031012, 22171282, 22261132511 and 22322111), Science and Technology Commission of Shanghai Municipality (21520780100, 22JC1401103 and 23JC1404500), Innovation Program of Shanghai Municipal Education Commission (2023ZKZD37), and the Youth Inno-vation Promotion Association of Chinese Academy of Sciences (Y2021075) for financial supports. S.L.Y. thanks the New Cornerstone Science Foundation for its support.

## Author contributions

S.-L.Y. conceived and supervised the project; M.Z. developed the [5 + 4] dearomative cycloaddition reactions of arenes; X.-L.H and M.L. con-tributed to expanding the scope of the reaction; Y.-J.G. and C.Z. per-formed the DFT calculations; C.Z. wrote the manuscript with the revisions suggested by all authors.

## Competing interests

The authors declare no competing interests.
