## [Peer Review File · Nature Communications]

Photo-induced intramolecular dearomative (5 + 4)
cycloaddition of arenes for the construction of highly strained
medium-sized-ringsREVIEWER COMMENTS

Reviewer #1 (Remarks to the Author):

Zheng, You, and co-workers report dearomative [5+4] cycloaddition of naphthalene derivatives with a tethered vinylcyclopropane under triplet sensitizing conditions. Several kinds of naphthalene derivatives were dearomatized to afford the corresponding products. DFT calculation provides a reasonable reaction pathway and explanation of the E-selectivity. Several transformation reactions of the product were demonstrated. The manuscript is clear and concise, and the supporting information is well-prepared.

The authors previously reported related dearomative cycloaddition reactions of indoles and pyrroles (ref. 35), and the key advancement of this work is the application of naphthalenes, which are less reactive than indoles and pyrroles, and the formation of a strained nine-membered ring containing a E-alkene moiety. However, the scope of this reaction is mostly limited to naphthalene derivatives containing a 2-carboalkoxy or 2-acyl functional group, and the obtained products are too special and specific to be used for the synthesis of medium-sized ring compounds discussed in the introduction. In terms of the scientific novelty, this work is a logical extension of ref. 35. Overall, the significance and novelty of this work would not meet the criteria required for Nature Communications.

Minor points

(1) In the introduction part, it would be better to cite and explain the reported works for medium-sized ring formations containing an E-alkene.

(2) Please add explanation of the limitations of the substrates. What happens if substrates without 2-substituent or with 2-alkyl group are used?

(3) For the DFT calculations, it would be generally recommended to use triple-zeta quality basis set, such as def2-TZVPP, for the final single point energy evaluation.

(4) Figure 2, legends: As DFT-D3 dispersion correction with BJ-damping is involved, it should be clarified.

Reviewer #2 (Remarks to the Author):

• Summary:

Significant Advance? Yes.

Quality & Clarity High.

Conclusions Supported? Mostly.

SI Document? Good (some data missing).

• Recommendation: Reconsider after (mostly minor) revisions.

• Main Comments: Zhu and co-workers report the discovery of a photosensitized intramolecular dearomative cycloaddition between arenes (mostly naphthol derivatives) and vinylcyclopropanes tethered to each other via an alkyl ether linkage. The transformation results not only in the dearomatization of the fused (or mononuclear) arenes, but also in the formation of densely functionalized 9-membered rings featuring an E-alkene, which is highly appealing. The use commercially available of visible-light photosensitizers increases the

practicality of the transformation. Although specific synthetic applications are not presented, various further transformations of the dearomatized (5+4) cycloadducts into potentially interesting polycyclic compounds are reported. Overall, this nice work eventually deserves to be published in a journal of the caliber of *Nat. Commun.* However, there are a few aspects of the manuscript that first warrant improvement: i) A key literature precedent by the P. J. Wagner group was omitted and must be discussed therein; ii) The authors are less than forthcoming about the limitations of the reported methods.

Our detailed comments follow below.

1) Perhaps this was missed by the authors as they also omitted it in their earlier work (ref. 35), but the reported cycloaddition is essentially a photosensitized version of the intramolecular arene-vinylcyclopropane previously reported by K.-L. Cheng and P. J. Wagner in 1994 (*J. Am. Chem. Soc.* 1994, 116, 7945; see also *Acc. Chem. Res.* 2001, 34, 1), who also noted the dearomatization with the formation of the E,Z-cyclononadiene containing polycyclic framework as the major product. In many regards, this manuscript by Zhu and co-workers appear to be the product of the fusion of the Glorius lab's photosensitized naphthyl ether-alkene photocycloaddition (ref. 46) with the earlier Wagner group report. In the earlier work, however, the substrate (an aromatic ketone) itself acted as the triplet photosensitizer, requiring UV irradiation. Furthermore, the manuscript was focused as a mechanistic investigation of a single substrate, rather than as a generally applicable cycloaddition method. The mechanistic interpretation of Cheng & Wagner also agrees with the one reported in this manuscript. Nevertheless, the advances contributed by the Zheng & You labs – notably identifying a suitable visible-light photosensitizer, exploring the substrate scope and further transformations – are significant. In any case, citations to the work of the Wagner lab must be added and fully discussed in the text. Preferably, both the examples of the Wagner lab (see above) and that of the Glorius lab (ref. 46) should be added to Scheme 1 to provide the reader with the proper context and background information.

2) We believe that the limitations of the (5+4) cycloaddition were either not investigated or not reported in a sufficient amount of detail. For instance:

- All the substrates reported in Table 2 feature a carbonyl group (ketone or ester) at the 2-position. Was that a requirement for the reaction to proceed (perhaps involving EnT or the proposed radical ring closure), or a only consequence of synthetic expediency? Were other electron-withdrawing groups (e.g. CN), or alkyl groups suitable? What about the parent 1-naphthol and 2-naphthol derivatives?

- All the substrates reported used the same alkyl ether linker between the naphthalene ring and the vinylcyclopropane group. The synthesis of the alkyl bromide precursor S1 should be amenable to the preparation of shorter or longer analogues. Have the cycloadditions that would lead to the 8- or 10-membered rings been attempted, and were they successful? The You lab also has on hand the corresponding carboxylic acid variants of the vinylcyclopropane linker (cf. ref. 35); have the corresponding naphthyl esters been tested in the (5+4) cycloaddition to yield 9-membered rings or others? Would the (5+4) cycloaddition work as well with N- or C-based linkers between the naphthalene ring and the vinylcyclopropane (e.g. naphthylamine derivatives)? In the case of C-based linkers, the synthetic method of Hammond and Xu (*Adv. Synth. Catal.* 2018, 360, 3667) could provide a rather practical route to such substrates.

- Given the authors' stated long-term goal of "dearomatization of plain aromatics", the

inclusion of the two phenyl ethers 9a-b at the end of the manuscript (almost as an afterthought) is surprising. In view of the generally greater difficulty in carrying out cycloadditions of benzene derivatives compared to fused aromatics of heteroarenes, this is a remarkable result, yet the discussion and the scope presented are minimal. The result is not even mentioned in the abstract or the conclusion, as if it was meant to be hidden, somehow. Perhaps did the authors wish to discuss it separately in a forthcoming full paper. Nevertheless, these raise additional questions: The cycloaddition of 9a-b was performed without a photosensitizer, using 370 nm irradiation (essentially replicating conditions from the Wagner report, albeit with an ortho rather than a para isomer, and following the initial cycloaddition by a further [2+2]). What happened to 9a-b in the presence of the photosensitizer? More importantly, what happened to the ordinary naphthyl ether substrates (e.g. 1a) upon 370 nm irradiation (or with deeper UV light, as opposed to the standard 455 nm blue LEDs) without the photosensitizer? These two questions should minimally be addressed & discussed in a revised manuscript.

• Other Comments / Suggestions / Corrections:

- Unless we are mistaken, the reaction should be described as (5+4) cycloaddition – within parentheses – instead of a [5+4] cycloaddition – within brackets. The former refers to the number of atoms involved (older notation) whereas the latter refers to the number of electrons involved (Woodward-Hoffman notation). See: <https://goldbook.iupac.org/terms/view/C01496> .

- The reaction appears to be rather slow: 24 h (monitored by TLC). Is a full day truly required to reach full consumption of the starting material, or were those not optimized? Performing a kinetic profile of the reaction mixture as a function of time would be a nice addition, and could provide information about the product distribution as a function of time (see other comment below).

- As the E/Z selectivity of the cycloaddition is emphasized as an important feature, some discussion of the following aspects would be appreciated:

i) How does the experimentally observed E/Z selectivity compare to that predicted by the DFT calculations? In other words, which is the selectivity-determining step of the mechanism based on the calculations and how does the energy difference compare with the empirical results? Is there a rationale for the lower E/Z selectivity observed for some substrates, such as for 1j => 2j?

ii) The dependence of the E/Z selectivity on the choice of photosensitizer appears counter-intuitive (i.e., despite high yields also obtained with PS V or VI). Based on the mechanism proposed from the DFT calculations in Figure 2, we cannot see how E/Z selectivity would be affected after sensitization (EnT) has taken place. Could it be that the composition of the mixture follows different kinetic profiles (isomerization of the initial product, etc.) depending on the choice of sensitizer? Acquiring a kinetic profile of the cycloaddition reaction mixture (see above) could perhaps help answer this question.

- If our understanding is correct, other mechanisms are ruled out on the basis of DFT calculations (though other mechanistic hypothesis have not been fully computed), literature precedents, and a single experiment (triplet quencher). I am inclined to believe that the authors are correct, though if one wanted to be thorough a Stern-Volmer quenching experiment and cyclic voltammetry performed on these substrates to rule out redox/ET pathways could provide stronger experimental support for the triplet sensitization hypothesis.

- It is probably beyond the scope of this paper, but one wonders what would happen if the cycloadduct 2 was treated with an olefin metathesis catalyst (with or without an added alkene) as a possible further transformation.
- P1, line 17: “photo-induced dearomative cycloadditions via energy transfer...” => “photo-induced dearomative cycloadditions that proceed via energy transfer...” (suggestion).
- P2, line 27: “It has been well known that...” => “It is well known that...” (suggestion)
- P3, line 51: “both Z- and E-olefin intermediates are generated...” => “both Z- and E-olefin intermediates were generated...”
- P5, line 84: “... which has relative small energy gaps...” => “... which have relatively small energy gaps...”
- P7, line 116: “Variation of the R1 group from acyl to...” => “Variation of the R1 group from acetyl to...”
- P13, line 201: “Discussion” => “Conclusion” [?]
- P9, lines 142-143: A citation to the relevant literature (evidence for Dexter energy transfer) should be added here.
- P9, lines 144-145: A citation to the relevant literature (triplet quencher) should be added here.
- Some of the references appear superfluous (or less relevant), and could be streamlined. For instance, refs. 58-69 may feature some over-indulgent self-citation. By contrast, the authors may consider Tanja Gaich’s chapter in *Comp. Org. Synth.* (10.1016/B978-0-08-097742-3.00516-4) to complement refs. 36-38. The title of ref. 39 appears to be missing.
- In the SI document, it would be preferable to cite the literature for the preparation of every substrate or precursor that is not commercially available. For example, several of the naphthol precursors A appear to have been made following the Glorius lab’s routes (ref. 46), though this has not been indicated. For precursors that are not literature compounds, the synthetic preps and full characterization should be added to the SI (precursors A to 1k, 1m, 1p, 1q, 1r, 1t, etc.).

Reviewer #3 (Remarks to the Author):

This manuscript describes the construction of highly strained medium-sized-rings through a photo-induced intramolecular dearomative [5+4] cycloaddition of arenes. In particular, this reaction provided unique polycyclic molecules with medium-sized-rings containing E-olefin. Further computational studies demonstrated the energy landscape of the reaction and the origin for the favorable formation of thermodynamically less stable products. Overall, I believe that this paper will provide readers with information on photo-induced dearomatization for the development of the construction of unique polycyclic molecules. However, here are several items that need to be addressed prior to publication (listed below). The following suggestions for revision and requests for additional data should be

considered by the author:

1) Page 4, line 59: The authors mention that “The dearomative cycloaddition reactions of benzene/naphthalene derivatives are much more challenging than those of heteroarenes...”. However, naphthalene, like other heterocycles, is considered to be highly reactive towards dearomative cycloaddition and should be distinguished from the benzene ring.

2) Page 4: It would be appropriate to cite Maestri's recent work in achieving intramolecular para-addition cyclization reactions promoted by visible light for simple aromatics. (Angew. Chem. Int. Ed. 2023, e202216817).

3) Table 1 and 2: E/Z ratios of substrates should be mentioned, as E/Z ratios of substrates might affect reactivity and the E/Z selectivity of the product.

4) Table 1: The authors should investigate the reactions of pure E-1a and Z-1a using the optimized condition, if possible.

5) Table 1: The authors should investigate the reactions with isolated pure products E-2a and Z-2a under the optimized condition to test whether there is conversion between E- and Z-products in the presence of photosensitizer under LED irradiation.

6) Figure 1: If possible, why not state the triplet energy of each catalyst in the Figure 1?

7) Table 2: All substrates have O-tether at C1 position and carbonyl group at C2 position. These substituents have significant effects on the reactivity, and thus their importance should be investigated. For examples, the authors should examine the reaction with the substrate with a C-tether and the substrate without a carbonyl group at C2-position.

8) The reaction with unsubstituted VCP (R3 = H) gave the desired reaction but the E/Z ratio decreased. Could you give any explanations for lower E/Z selectivity?

9) The authors should add the corresponding reference to the sentence “On the basis of our previous works and literature reports, the current dearomative [5 + 4] cycloaddition reaction is believed to proceed via a Dexter energy transfer mechanism” (Page 9, line 142).

10) Scheme 2: Regarding the reactions with 9a and 9b, why did not the authors use photosensitizer? What happens in these reactions when using photosensitizer?

Below please find our point-by-point responses to the reviewers' comments

Critical comments from Reviewer 1

1. In the introduction part, it would be better to cite and explain the reported works for medium-sized ring formations containing an *E*-alkene.

Our reply: We appreciate this suggestion. In the revised manuscript, we have added a statement on the traditional synthesis of *trans*-cyclononene and *trans*-cyclodecene derivatives in the main text (Page 2) and cited relevant references (Refs. 21 and 22) in the revised manuscript.

“Traditional methods to access these structures mainly relied on intramolecular cyclization, or the functional group manipulations on a pre-assembled medium-sized-ring (such as elimination, semi-hydrogenation, and photoinduced isomerization, *etc.*), which usually suffered from low reactivity and selectivity issues.”

2. Please add explanation of the limitations of the substrates. What happens if substrates without 2-substituent or with 2-alkyl group are used?

Our reply: We appreciate this suggestion. As suggested, we have tried the reactions of naphthalene-derivatives with R = H or Me. However, under the modified conditions, the reactions did not occur but with starting materials recovered. We have listed the unsuccessful substrates in the revised SI (Page S190).

3. For the DFT calculations, it would be generally recommended to use triple-zeta quality basis set, such as def2-TZVPP, for the final single point energy evaluation?

Our reply: We appreciate this suggestion. We have performed the single-point calculations at the (U)M062X-D3/def2-TZVPP (SMD, DCM) level of theory, and the energy values of all calculated species have updated in the revised manuscript (Figure 2 and Pages 11–12). The computational results have been added to the revised SI (Pages S39–S65)

4. Figure 2, legends: As DFT-D3 dispersion correction with BJ-damping is involved, it should be clarified.

Our reply: Agreed. The legend of **Figure 2** has been modified in the revised manuscript so that the information of dispersion correction is included (**Page 11**).

Critical comments from Reviewer 2

1. In any case, citations to the work of the Wagner lab must be added and fully discussed in the text. Preferably, both the examples of the Wagner lab (see above) and that of the Glorius lab (ref. 46) should be added to Scheme 1 to provide the reader with the proper context and background information.

Our reply: We appreciate this suggestion. In the revised manuscript, we have added the discussion on the work of Wagner (Page 2) and cited it as Ref. 23.

“To be noted, Cheng and Wagner reported in 1994 an intramolecular cycloaddition of cyclopropyl-substituted *p*-(butenyloxy)acetophenone under the irradiation of ultraviolet (UV) light. The (5 + 4) adduct possessing an *E*-olefin was identified as a major product in the reaction mixture, but not isolable due to its instability.”

In addition, we have redrawn Scheme 1 by replacing the reactions in the original panels b and c with the works of Wagner (Ref. 23) and Glorius (Ref. 50, renumbered).

2. All the substrates reported in Table 2 feature a carbonyl group (ketone or ester) at the 2-position. Was that a requirement for the reaction to proceed (perhaps involving EnT or the proposed radical ring closure), or an only consequence of synthetic expediency? Were other electron-withdrawing groups (e.g. CN), or alkyl groups suitable? What about the parent 1-naphthol and 2-naphthol derivatives?

Our reply: We appreciate this suggestion. We have tried the reactions of naphthalene-derivatives with R = H or Me. However, under the modified conditions, the reactions did not occur but with starting materials recovered.

Besides, our synthetic procedure for the substrates (O-acylation and Fries rearrangement) did not allow the introduction of a CN group to the naphthalene ring. According to our previous knowledge on the structure-reactivity relationship for indole-derivatives in similar energy-transfer mediated dearomative cycloaddition reactions, introducing an electron-withdrawing group is beneficial for the reactivity. We have added a brief discussion on this issue to the revised manuscript (Page 9).

“Notably, if the R¹ group was switched to methyl or H, the desired reaction was not observed with the starting materials recovered. According to our previous works,

adding an electron-withdrawing group would be beneficial for the energy-transfer mediated dearomative cycloaddition.”

We have listed the unsuccessful substrates in the revised SI (Page S190).

3. The synthesis of the alkyl bromide precursor S1 should be amenable to the preparation of shorter or longer analogues. Have the cycloadditions that would lead to the 8- or 10-membered rings been attempted, and were they successful? The You lab also has on hand the corresponding carboxylic acid variants of the vinylcyclopropane linker (cf. ref. 35); have the corresponding naphthyl esters been tested in the (5+4) cycloaddition to yield 9-membered rings or others? Would the (5+4) cycloaddition work as well with N- or C-based linkers between the naphthalene ring and the vinylcyclopropane (e.g. naphthylamine derivatives)? In the case of C-based linkers, the synthetic method of Hammond and Xu (Adv. Synth. Catal. 2018, 360, 3667) could provide a rather practical route to such substrates.

Our reply: We appreciate this suggestion. First, it should be pointed out that manipulating the linkage between the Br atom and the olefin of the VCP moiety (either removing the methylene group or adding one) will not change the nine-membered ring to eight- or ten-membered ones, but only alter the spiro ring structure (See the scheme below).

Indeed, we have tried to synthesize the cycloadducts of diverse structures. For example, we have prepared compound S2 bearing a conjugate diene attached to cyclopropane, aiming for an eleven-membered ring via (7 + 4) cycloaddition. However, after purification, S2 quickly underwent polymerization to afford some jelly-like polymers (monitored by ¹H NMR). Thus, attempt for this reaction was failed (See the scheme below).

We have also tried the substrates with an ester or amide linkage. However, all these attempts were failed either due to the inaccessibility of the desired substrate or no target reaction happened (See the scheme below).

For the substrate with C-based linkage, we have checked the reference mentioned by this reviewer (*Adv. Synth. Catal.* **2018**, 360, 3667.), but found that both the required coupling partners are unknown compounds that are quite challenging to synthesize. Thus, we decide not to pursue this reaction in the current work.

We have listed the unsuccessful substrates in the revised SI (Page S190).

4. The cycloaddition of 9a-b was performed without a photosensitizer, using 370 nm irradiation (essentially replicating conditions from the Wagner report, albeit with

an ortho rather than a para isomer, and following the initial cycloaddition by a further [2+2]). What happened to 9a-b in the presence of the photosensitizer? More importantly, what happened to the ordinary naphthyl ether substrates (e.g. 1a) upon 370 nm irradiation (or with deeper UV light, as opposed to the standard 455 nm blue LEDs)) without the photosensitizer? These two questions should minimally be addressed & discussed in a revised manuscript.

Our reply: We appreciate this suggestion. We have tried the reactions suggested by this reviewer. In the presence of blue LEDs and the photosensitizer used in this study, no reaction was occurred with benzene-derived substrate **9a** (all starting materials recovered). However, under the irradiation of UV light (370 nm), standard substrate **1a** underwent cascade (5 + 4)/(2 + 2) cycloaddition, leading to a mixture of **7** and **8** in 39% and 36% yield, respectively. We have added the results of these reactions in **Scheme 2** and associated text of the revised manuscript (**Page 14**).

“It should be mentioned that under the irradiation of UV light (370 nm), compound **1a** could be converted to a mixture of **7** (39% yield) and **8** (36% yield), and benzene-derived VCPs **9a** and **9b** to [2.2.0] bicyclic analogs **10a** (48% yield) and **10b** (30% yield), respectively. Besides, **9a** remained intact under blue LEDs with photosensitizer **VI** or thioxanthone.”

5. Unless we are mistaken, the reaction should be described as (5+4) cycloaddition – within parentheses – instead of a (5+4) cycloaddition – within brackets. The former refers to the number of atoms involved (older notation) whereas the latter refers to the number of electrons involved (Woodward-Hoffman notation). See: <https://goldbook.iupac.org/terms/view/C01496>.

Our reply: Agreed. We have changed the brackets to parentheses in the notation of cycloaddition reactions throughout the revised manuscript and SI.

6. The reaction appears to be rather slow: 24 h (monitored by TLC). Is a full day truly required to reach full consumption of the starting material, or were those not optimized? Performing a kinetic profile of the reaction mixture as a function of time would be a nice addition, and could provide information about the product distribution as a function of time (see other comment below).

Our reply: We appreciate this suggestion. Actually, the time required for the completion of reaction could be varied according to the substrates. The purpose for setting the reaction time to 24 h was to guarantee the full conversion of some special substrates (like **1j**). Since the desired products were stable under the standard conditions, this setting should have minimal influences to the reaction outcomes. The major aim of this work is the discovery and development of an energy-transfer mediated cycloaddition reaction toward highly strained medium-sized-rings. Acquiring reliable kinetic profile of the reaction experimentally will be a subject of our future studies.

7. How does the experimentally observed *E/Z* selectivity compare to that predicted by the DFT calculations? In other words, which is the selectivity-determining step of the mechanism based on the calculations and how does the energy difference compare with the empirical results? Is there a rationale for the lower *E/Z* selectivity observed for some substrates, such as for **1j** => **2j**?

Our reply: We appreciated this suggestion. Based on our computational studies, we believe that the *E/Z* selectivity of the product should be stemmed from the step of ring-opening of cyclopropane ring (**TS2-E-T₁** vs. **TS2-Z-T₁**). After the refinement of the calculated energies as suggested by Reviewer #1, the difference of the Gibbs free energies of these two transition states is 0.9 kcal mol⁻¹, which well agrees with the experimentally observed selectivity, 7.7:1 for *E-2a/Z-2a* at rt. For the formation of *E-2j* and *Z-2j*, we believe the removal of *gem*-diester groups would attenuate the energetic difference between the two transition states for ring-opening due to the missing of steric congestion between the *gem*-diester groups and the naphthalene ring. Thus, a lower ratio of 2.6:1 should be acceptable. We have modified our statement on the origin of selectivity in the revised manuscript to make it clearer (Pages 12 and 13).

“The discrimination between the two reaction pathways was originated during the formation of the *E*- or *Z*-olefin in the step of ring-opening of cyclopropane ring. The formation of *E*-olefin via the transition state **TS2-E-T₁** was favorable than **TS2-Z-T₁** that led to *Z*-olefin. The lower *E/Z* ratio of **2j** (2.6/1) might be attributed to the attenuated energetic difference between the transition states for ring-opening of cyclopropane ring due to the missing of steric congestion between the *gem*-diester groups and the naphthalene ring.”

8. The dependence of the *E/Z* selectivity on the choice of photosensitizer appears counter-intuitive (i.e., despite high yields also obtained with PS V or VI). Based on the mechanism proposed from the DFT calculations in Figure 2, we cannot see how *E/Z* selectivity would be affected after sensitization (EnT) has taken place. Could it be that the composition of the mixture follows different kinetic profiles (isomerization of the initial product, etc.) depending on the choice of sensitizer? Acquiring a kinetic profile of the cycloaddition reaction mixture (see above) could perhaps help answer this question.

Our reply: We appreciate this suggestion. Indeed, during the optimization of reaction conditions, it was found that the *E/Z* ratio largely dropped (from 7.7/1 to less than 3/1) when photosensitizers **V** and **VI** (with higher triplet energies) were utilized, respectively (entries 4–6, Table 1). The most probable reason behind this phenomenon is the less stability of *E-2a* compared with that of *Z-2a* in the presence of **V** or **VI**. We have checked the stability of *E-2a* under **V** with ¹H NMR, and found the gradual transformation of *E-2a* to *Z-2a* along with the cycloadducts **7** and **8** (See the figure below).

We have added a brief discussion on this issue in the revised manuscript (Page 6) and SI (Pages S37 and S38).

“The lower *E/Z* ratios (less than 3/1, entries 5 and 6) with photosensitizers **V** or **VI** were attributed to the conversion of *E-2a* to *Z-2a* along with further (2 + 2) cycloadducts **7** and **8** (*vide infra*) under these conditions.”

9. It is probably beyond the scope of this paper, but one wonders what would happen if the cycloadduct **2** was treated with an olefin metathesis catalyst (with or without an added alkene) as a possible further transformation.

Our reply: We have included a series of transformations of *E-2a* in this work (Scheme 2). As agreed by this reviewer, the ring-opening metathesis of this compound will be a subject of our future studies.

10. P1, line 17: “photo-induced dearomative cycloadditions via energy transfer” => “photo-induced dearomative cycloadditions that proceed via energy transfer” (suggestion).

Our reply: Done. This change has been updated in the revised manuscript (Page 1).

11. P2, line 27: “It has been well known that” => “It is well known that” (suggestion).

Our reply: Done. This change has been updated in the revised manuscript (Page 2).

12. P3, line 51: “both Z- and E-olefin intermediates are generated” => “both Z- and E-olefin intermediates were generated”.

Our reply: Done. This change has been updated in the revised manuscript (Page 3).

13. “which has relative small energy gaps” => “which have relatively small energy gaps”.

Our reply: Done. This change has been updated in the revised manuscript (Page 5).

14. P7, line 116: “Variation of the R1 group from acyl to” => “Variation of the R1 group from acetyl to”.

Our reply: Done. This change has been updated in the revised manuscript (Page 8).

15. P13, line 201: “Discussion” => “Conclusion” [?].

Our reply: According to the format requirement of *Nature Communication*, the main text is usually closed with a section under the subheading “Discussion”. We prefer to leave the decision on this issue to the Editor.

16. P9, lines 142-143: A citation to the relevant literature (evidence for Dexter energy transfer) should be added here.

Our reply: Done. We have cited recent reviews on dearomative cycloaddition reactions mediated by Dexter energy transfer explicitly as Refs. 24–27 in the revised manuscript.

17. P9, lines 144-145: A citation to the relevant literature (triplet quencher) should be added here.

Our reply: Done. The literature about this triplet quencher was cited as Ref. 70 in the revised manuscript.

18. Some of the references appear superfluous (or less relevant), and could be streamlined. For instance, refs. 58-69 may feature some over-indulgent self-citation. By contrast, the authors may consider Tanja Gaich's chapter in *Comp. Org. Synth.* (10.1016/B978-0-08-097742-3.00516-4) to complement refs. 36-38. The title of ref. 39 appears to be missing.

Our reply: Done. The *Comp. Org. Synth.* paper has been cited as **Ref. 41** in the revised manuscript. The citation of reviews on general dearomatization reactions has been modified to keep a relevant and balanced view (**Refs. 63–69**).

19. In the SI document, it would be preferable to cite the literature for the preparation of every substrate or precursor that is not commercially available. For example, several of the naphthol precursors A appear to have been made following the Glorius lab's routes (ref. 46), though this has not been indicated. For precursors that are not literature compounds, the synthetic preps and full characterization should be added to the SI (precursors A to **1k**, **1m**, **1p**, **1q**, **1r**, **1t**, etc.).

Our reply: We appreciate this suggestion. We have cited the work from the Glorius group in the revised SI as **Ref. 2** (The following citations have been renumbered). As suggested, the procedures for substrates **1k**, **1m-1p**, **1q**, **1r** and **1t** have been added in the revised SI (**Pages S4 to S7**).

Critical comments from Reviewer 3

1. Page 4, line 59: The authors mention that “The dearomative cycloaddition reactions of benzene/naphthalene derivatives are much more challenging than those of heteroarenes...”. However, naphthalene, like other heterocycles, is considered to be highly reactive towards dearomative cycloaddition and should be distinguished from the benzene ring.

Our reply: We appreciate this suggestion. This statement has been modified as the follows in the revised manuscript (Page 4).

“The dearomative cycloaddition reactions of electronically unbiased arenes are more challenging than ...”

2. Page 4: It would be appropriate to cite Maestri's recent work in achieving intramolecular para-addition cyclization reactions promoted by visible light for simple aromatics. (Angew. Chem. Int. Ed. 2023, e202216817).

Our reply: Done. This paper has been cited in the revised manuscript as Ref. 61.

3. Table 1 and 2: E/Z ratios of substrates should be mentioned, as E/Z ratios of substrates might affect reactivity and the E/Z selectivity of the product.

Our reply: Done. The E/Z ratio of **1a** (4/1) has been added to Table 1 and the associated text in the revised manuscript (Page 5). The E/Z ratios of other substrates (**1b–1t**, **9a** and **9b**) have all been mentioned in the SI (Pages S8 to S18)

4. Table 1: The authors should investigate the reactions of pure E-1a and Z-1a using the optimized condition, if possible.

Our reply: We appreciate this suggestion. Actually, we have performed the reaction using **1a** with different E/Z ratios (prepared from different batches), the same E/Z ratios of products **2a** was obtained. In fact, mechanistically, the E/Z ratio of **2a** should be not relevant to that of **1a**, because after the generation of triplet 1,4-biradical via the first C–C bond-formation, the E/Z configuration in the substrate vanished, while the E/Z configuration in the product will only be set up during the subsequent ring-opening of cyclopropane ring (See the scheme below). Thus, we are quite confident that the same outcome in terms of E/Z ratio of **2a** will be obtained no matter whether pure E-**1a** or Z-**1a** is employed.

We have added a brief discussion on this issue in the revised manuscript (Page 13).

“The olefinic geometry of the substrate would not affect the *E/Z* ratio of the product since configurational difference of the substrate would vanish after the generation of triplet 1,4-biradical via the first C–C bond-formation, while the *E/Z* configuration in the product would be set up during the subsequent ring-opening of cyclopropane ring.”

- Table 1: The authors should investigate the reactions with isolated pure products *E-2a* and *Z-2a* under the optimized condition to test whether there is conversion between *E*- and *Z*-products in the presence of photosensitizer under LED irradiation.

Our reply: We appreciate this suggestion. As mentioned in our reply to Reviewer #2, we have checked the stability of *E-2a* under different photosensitizers. Under the standard condition (with photosensitizer **IV**), no isomerization of *E-2a* to *Z-2a* was observed. On the other hand, in the presence of photosensitizer **V**, gradual transformation of *E-2a* to *Z-2a* along with the cycloadducts **7** and **8** was observed by ¹H NMR (See the figure below).

We have added a brief discussion on this issue in the revised manuscript (Page 6) and SI (Pages S37 to S38).

“The lower *E/Z* ratios (less than 3/1, entries 5 and 6) with photosensitizers **V** or **VI** were attributed to the conversion of *E*-2a to *Z*-2a along with further (2 + 2) cycloadducts **7** and **8** (*vide infra*) under these conditions.”

6. Figure 1: If possible, why not state the triplet energy of each catalyst in the Figure 1?

Our reply: Done. We have added this information in Figure 1 of the revised manuscript.

7. Table 2: All substrates have O-tether at C1 position and carbonyl group at C2 position. These substituents have significant effects on the reactivity, and thus their importance should be investigated. For examples, the authors should examine the reaction with the substrate with a C-tether and the substrate without a carbonyl group at C2-position.

Our reply: We appreciate this suggestion. As mentioned in our reply to Reviewer #2, we have tried the substrates with an ester or amide linkage. However, all these attempts were failed either due to the inaccessibility of the desired substrate or no target reaction happened (See the scheme below).

For the substrate with C-based linkage, we have checked the reference mentioned by Reviewer #2 (*Adv. Synth. Catal.* **2018**, 360, 3667.), but found that both key coupling partners are unknown compounds that are quite difficult to synthesize. Thus, we decide not to pursue this reaction in the current work.

Besides, we have tried the reactions of naphthalene-derivatives with R = H or Me. However, under the modified conditions, the reactions did not occur but with starting materials recovered.

We have listed all the unsuccessful substrates in the revised SI (Page S190).

8. The reaction with unsubstituted VCP ($R_3 = H$) gave the desired reaction but the *E/Z* ratio decreased. Could you give any explanations for lower *E/Z* selectivity?

Our reply: We appreciate this suggestion. Based on our DFT calculations, we believe the *E/Z* ratios of the products should be determined in the step of ring-opening of cyclopropane ring. The difference of the calculated Gibbs free energies of these two transition states (**TS2-*E*-T₁** and **TS2-*Z*-T₁**) is 0.9 kcal mol⁻¹, which well agrees with the experimentally observed selectivity, 7.7:1 for *E*-**2a**/*Z*-**2a** at rt. For the formation of *E*-**2j** and *Z*-**2j**, we believe the removal of *gem*-diester groups would attenuate the energetic difference between the two transition states for ring-opening due to the missing of steric congestion between the *gem*-diester groups and the naphthalene ring. Thus, a lower ratio of 2.6:1 should be acceptable. We have modified our statement on the origin of selectivity in the revised manuscript to make it clearer (Page 13).

“... the discrimination between the two reaction pathways was originated during the formation of the *E*- or *Z*-olefin in the step of ring-opening of cyclopropane ring. The formation of *E*-olefin via the transition state **TS2-*E*-T₁** was favorable than **TS2-*Z*-T₁** that led to *Z*-olefin. The lower *E/Z* ratio of **2j** (2.6/1) might be attributed to the attenuated energetic difference between the transition states for ring-opening of cyclopropane ring due to the missing of steric congestion between the *gem*-diester groups and the naphthalene ring.”

9. The authors should add the corresponding reference to the sentence “On the basis of our previous works and literature reports, the current dearomative (5+4) cycloaddition reaction is believed to proceed via a Dexter energy transfer mechanism” (Page 9, line 142).

Our reply: Done. We have cited recent reviews on dearomative cycloaddition reactions mediated by Dexter energy transfer explicitly as Refs. 24–28 in the revised manuscript.

10. Scheme 2: Regarding the reactions with **9a** and **9b**, why did not the authors use photosensitizer? What happens in these reactions when using photosensitizer?

Our reply: We appreciate this suggestion. We have tried the reactions suggested by this reviewer. In the presence of blue LEDs and the photosensitizer used this study, no reaction occurred with benzene-derived substrate **9a** (all starting materials recovered). We have added the results of these reactions in Scheme 2 and associated text of the revised manuscript (Pages 13 and 14).

“It should be mentioned that under the irradiation of UV light (370 nm), compound **1a** could be converted to a mixture of **7** (39% yield) and **8** (36% yield), and benzene-

derived VCPs **9a** and **9b** to [2.2.0] bicyclic analogs **10a** (48% yield) and **10b** (30% yield), respectively. Besides, **9a** remained intact under blue LEDs with photosensitizer **VI** or thioxanthenone.”

REVIEWERS' COMMENTS

Reviewer #1 (Remarks to the Author):

The raised issues have been appropriately addressed. In terms of the novelty and significance of the work, my opinion has not been fully changed but I appreciate the other reviewers' comments and the decision of the editor.

Reviewer #2 (Remarks to the Author):

We are satisfied with the revised version. We now recommend this manuscript for publication.

Reviewer #3 (Remarks to the Author):

The authors have addressed most of my comments and suggested corrections. However, there is one minor comment. The authors list unsuccessful substrates in the Supplementary Information (Page S190). Since this information is valuable for the reader, it is recommended that a note be included in the main text indicating that these results are listed in the Supplementary Information.

Below please find our point-by-point responses to the reviewers' comments

Critical comments from Reviewer 1

1. The raised issues have been appropriately addressed. In terms of the novelty and significance of the work, my opinion has not been fully changed but I appreciate the other reviewers' comments and the decision of the editor.

Our reply: We appreciate this comment.

Critical comments from Reviewer 2

1. We are satisfied with the revised version. We now recommend this manuscript for publication.

Our reply: We appreciate this comment.

Critical comments from Reviewer 3

1. The authors have addressed most of my comments and suggested corrections. However, there is one minor comment. The authors list unsuccessful substrates in the Supplementary Information (Page S190). Since this information is valuable for the reader, it is recommended that a note be included in the main text indicating that these results are listed in the Supplementary Information.

Our reply: We appreciate this suggestion. We have added one note in the main text indicating this list in the Supplementary Information (**Page 6**).